# Encourage or inhibit: A study on the impact of corporate digital transformation on management's tone manipulation of information disclosure

**Lingyun Yang[1], Zhihong Zhang[1]\*, Lulu Wang[2], Yikai Liang[3]**

**1** School of Accountancy, Shandong University of Finance and Economics, Jinan, Shandong, China,
**2** School of Vocational Education Shandong, Youth University Of Political Science, **3** School of Management Science and Engineering, Shandong University of Finance and Economics, Jinan, Shandong, People's Republic of China

\* jqbai@126.com

## Abstract

Digitalization is anticipated to substantially improve information transparency and enhance the capital market's information environment. However, during the initial phase of digital transformation, companies face challenges in achieving high-quality information disclosure. This is because digital technology implementation and organizational adjustment are still at early stages. Concurrently, investors face difficulties in assessing the accuracy of disclosed information. This challenge provides management with the opportunity to overstate the benefits of digital transformation. This study investigates the impact of corporate digital transformation on management's tone manipulation behavior, using a sample of Chinese A-share listed companies from 2012 to 2021. Findings indicate that corporate digital transformation significantly fosters management's tone manipulation during its exploration phase. Media slant positively moderates this relationship. Further analysis supports the paper's hypothesis: companies with weaker financial flexibility and lower risk information disclosure levels show a stronger positive correlation between digital transformation and tone manipulation. Concurrently, mechanism analysis reveals that management overconfidence partially mediates the relationship. This suggests that digital transformation increases managerial overconfidence, thereby promoting tone manipulation. The conclusion offers new insights into enhancing management discussion and analysis information disclosure quality from a corporate strategic transformation perspective. It serves as a valuable reference for accurately identifying misleading management signals.

## 1. Introduction

The rapid emergence of new technologies, business models, and platforms has made cultivating digital competitiveness a global focus. The "Global Computing Power Index Assessment Report" is jointly compiled by Tsinghua University and other institutions. It indicates despite a general slowdown in global GDP growth, the digital economy remains robust. The share

**Data availability statement:** The annual reports of Chinese listed companies can be downloaded from the company's official website, which is publicly available free of charge. Individuals who want to access these data can contact CSMAR for the data (see https://data.csmar.com for more details, contact via 400-639-888), Wind (see https://www.wind.com/ for more details, contact via 400-820-9463), CNRDS (see https://www.cnrds.com, contact via 021-66181082). For example, if you want to download data related to enterprise-level variables, you can log in the CSMAR database, select "Financial Statement Library" from "Company Research Series" and choose to download the required information based on the variable measure. We confirm that others would be able to access these data in the same manner as the authors.

**Funding:** (1) Key Research and Development Program (soft science) project of Shandong Province, China, grant number [2024RZA0201]. (funder: Zhihong Zhang) (2) The Humanities and Social Sciences Project of the Ministry of Education of China, grant number [22YJA630118]. (funder: Zhihong Zhang) Zhihong Zhang (ZZ) contribution statement: Conceptualization, Validation, Writing - Review & Editing, Supervision, Funding acquisition, Project administration.

**Competing interests:** The authors have declared that no competing interests exist.

of the digital economy in the GDP of major countries worldwide continues to rise and is expected to reach 54% by 2026. The digital transformation of enterprises is the core driver and foundation for the development of the digital economy. In response to market uncertainties and significant shifts in consumer demands, digitalization is increasingly integrating into the value chains of enterprises. Digital tools and platforms facilitate this integration. It is transforming products, processes, services, and organizational structures. This marks digital transformation as a strategic choice for many enterprises aiming to boost performance, innovation, management reform, and competitive advantage [1–6]. It also enhances industrial efficiency and facilitates the upgrading of industrial [7–10]. However, despite significant investments in digital operations and infrastructure, not all enterprises have seen the desired outcomes [11]. A mismatch often exists between an enterprise's management systems and the advanced technological frameworks of digital transformation. This mismatch, along with the continuous investment in digital elements, can lead to an "IT paradox" [12]. It means that not every enterprise benefits from digital transformation, yet all face varying degrees of challenges and risks [11]. According to Accenture's China Digital Transformation Index 2022, only 17% of Chinese enterprises currently experience significant benefits from digital transformation. This indicates that the digital transformation of Chinese enterprises is still in an exploratory phase, with immature strategic systems. Only a handful of enterprises have successfully leveraged digital technology for comprehensive, end-to-end interconnectivity and dynamic configuration both internally and externally. However, "Digital transformation" represents a positive signal to investors. Companies are often eager to share such beneficial news, highlighting the competitive advantage it brings. This is called impression management. Impression management refers to the process where people are often concerned about how others evaluate them, both consciously and unconsciously. They have a subconscious desire to be viewed positively [13,14]. During interactions, individuals actively manage and construct their positive image to influence others' perceptions. This can lead to recognition and the establishment of a positive reputation, aiding in self-fulfillment and economic benefits [15,16]. Viewing through the lens of information manipulation, management is motivated to craft a positive image by exaggerating digital transformation benefits. This approach helps guide public views on the company's development and diminish negative evaluations. Additionally, it enhances attention and financial support from social groups.

Signaling theory posits that signals are transmitted from those with an information advantage to those at a disadvantage [17]. This theory in management explains how companies use signaling to address information asymmetry in exchanges. Companies, holding an informational advantage, disclose business and financial details to the public. This helps alleviate information asymmetry and diminish adverse selection [18,19]. However, driven by self-interest, management often uses information asymmetry to strategically disclose non-accurate information. They may also manipulate disclosure methods, including tone manipulation. The corporate annual report serves as the core of management's information disclosure system. It acts as a key communication bridge. The Management Discussion and Analysis (MD&A) section serves as a primary source for conveying qualitative information in a company's annual report. It is characterized by its predictive and narrative nature. This allows for subjective and flexible information disclosure. It not only addresses investors' needs for information on business conditions, development plans, and future risks [20,21], but also serves the critical role of helping management to communicate clearly with investors [22,23]. Research indicates that enterprises' qualitative information disclosures offer valuable incremental insights, influencing user decision-making [24]. The Chinese language's nuanced and indirect nature and the broader latitude in textual information provide management with greater leeway. This leeway is particularly evident in tone management within the Chinese

capital market. Both positive and negative tones carry substantial information, significantly impacting the market [25]. Furthermore, most Chinese stock market investors are individuals. This makes accurately assessing corporate disclosures a significant challenge. This offers management opportunities to manipulate information. Therefore, this study examines MD&A disclosures by A-share listed companies from 2012 to 2021, focusing on text tone to empirically assess how digital transformation affects tone manipulation.

Compared with existing literature, this paper's contribution is: First, few studies explore the relationship between digital transformation and information disclosure. Even fewer investigate the intermediary role of management information disclosure behavior. This paper provides evidence for the impact of corporate digital transformation on management tone manipulation. This expands the scope of research related to digital transformation. Second, it enriches the research on factors influencing management information disclosure. Research has investigated the impact of various factors on strategic information disclosure. These factors include a company's financial situation [26], the level of competition in the product market [27], corporate transactions [28,29], expected future performance [30] and so on. By employing signaling theory and impression management, this paper focuses on how corporate digital transformation impacts management's tone manipulation in the digital era. It provides new perspectives for interpreting textual information in annual reports. This, in turn, deepens the research related to information manipulation. Third, this paper has practical value and implications. Corporate digital transformation increases the positivity of management's tone, but these positive signals lack credibility and are results of impression management. This assists investors in better interpreting the actual information content in MD&A and identifying misleading management signals. It also offers regulators insights to identify opportunistic behaviors of management. Additionally, it helps them regulate and monitor text information disclosure from both internal and external perspectives of the company.

The structure design of this paper is as follows: Section 1 is an introduction. It outlines the current development status of enterprises during the digital transformation exploration period. Additionally, it examines the relationship between digital transformation and management's strategic information disclosure. Section 2 presents the literature review and hypotheses development. It highlights the contributions and shortcomings of relevant research. And it delves into the impact mechanism of digital transformation on management's tone manipulation and puts forward hypotheses. Section 3 is research design, including data, variable constructions, model design and research samples. Section 4 analyzes empirical regression results and conducts robustness tests. In Section 5, further analysis is conducted on the motivations and mechanism analysis behind management's tone manipulation in digitally transforming enterprises. Finally, Sections 6 summarize the main work, contributions, and limitations of this study.

## 2. Literature review and hypotheses development

### 2.1. Literature review of tone manipulation

Management tone manipulation involves using power to increase positive wording and decrease negative expressions in qualitative statements. This strategy enhances the overall tone, making the disclosed information more optimistic [29]. In recent years, research on strategic management disclosure have focused on annual report/10-K/10-Q filings [31–33], media news [34], earnings press releases [29,35], analyst reports [36], and conference calls [37,38].

Starting from the agency theory and incomplete contract theory [39,40], existing studies elaborate on the function of textual tone from two different perspectives. The first perspective is the incremental information view. Managers reveal private information about the

company's current operations and future performance, which not fully captured by financial metrics. By conveying thus information, they distinguish their firms from others [41,42]. The second perspective focuses on information manipulation. As investors are the key recipients of corporate disclosures, the management usually highlight positive business developments to reduce negative investor reactions. Given the limited attention hypothesis, investors' constrained time and cognitive resources make it challenging to fully focus on all disclosed information [43]. This motivates management to leverage their informational advantage for impression management. They may manipulate disclosures to mask poor performance or inflate prospects, aiming to influence investor sentiment. Misleading information that does not align with facts can distort investor judgment and decisions [27]. It often serves specific objectives [44,45].

The asymmetric information game theory suggests that agents with an informational advantage engage in strategic behavior. They optimize their interests through exploiting information asymmetry [46]. Managers manipulate tone for impression management to gain stock market profits, improve bond credit ratings, and stabilize their reputation and position [29,47,48]. A higher degree of tone manipulation increases the likelihood of a company engaging in earnings management [44,45]. It also raises the possibility of using misleading information to achieve higher credit ratings [48]. These actions not only disregard the principle of accounting prudence but also increase the risk of stock price crashes [35,49]. Meanwhile, scholars explore the governance effects of management tone manipulation through various lenses, such as product market [27], industry competition [50], capital markets and legal environments [51], audit committee characteristics [33], and CEO characteristics [52].

Digital transformation facilitates a comprehensive strategic shift towards data-driven management and innovation [1,53–57]. Research into how digital transformation affects management's tone manipulation is nascent. Nevertheless, existing studies on the impact of digital transformation on information disclosure have showed varied conclusions. From the perspective of information disclosure channels, various platforms facilitate the deep integration of the digital and real economies. Internet platforms create a positive interactive communication environment. Enterprises use these platforms as vehicles to efficiently transmit operational information to users [58]. For example, companies can promote and sell products through social media platforms like Weibo, Facebook, and TikTok. Xiaomi represents a successful example of using social media to disrupt traditional startup models. Another example is the interactive platforms launched by the Shenzhen and Shanghai Stock Exchanges. These platforms facilitate real-time communication. They provide cost-effective and efficient communication channels. These channels particularly benefiting small and medium investors. This approach significantly enhances the speed and fairness of acquiring information. However, internet platforms may also support management's concealment activities. They can serve as channels for manipulative information disclosure [33,59,60]. Such strategic disclosure practices can exacerbate market information asymmetry. This form of improper competition lowers the efficiency of information matching [61,62]. From the perspective of information disclosure quality, digital transformation can improve internal control effectiveness by alleviating information asymmetry [63]. This process enhances the transparency of internal control information and elevates the level of its disclosure. Externally, it can also improve the accuracy of corporate information disclosure [64,65] or promote herding behavior in corporate ESG information disclosure [66]. However, digital transformation's promotion of green information disclosure might merely reflect superficial compliance by companies [67]. That is, the disclosed information might also include strategic concept speculation behavior [60]. Typically, in the digital economy, companies actively disclose more digital transformation information to signal innovation and competitiveness. However, these disclosures might be

more talk than action, merely a tactic of conceptual hype [68,69]. This exaggeration can lead information users to opportunistic behavior, influencing their judgment and decisions. In practice, China's securities regulators have identified instances of misleading disclosures on popular topics. These disclosures confuse investors and lead to stock price manipulation. For example, the CSRC penalized companies like Bright Oceans Inter-Telecom Corporation and Jiangsu Daybright Intelligent Electric Co., Ltd. These companies ambiguously disclosed digital technology applications, such as AI and big data, in reports and interactive platforms. These companies eventually supplemented their disclosures of related risks under regulatory pressure. Therefore, in the current digitalization era, the effects of digital transformation on the capital market's information environment are still under investigation. And the specific impact on corporate information disclosure is still unclear. In particular, the effects on management's strategic disclosure practices remain an open question.

## 2.2. Digital transformation and tone manipulation

Corporate governance research is inherently linked to socio-economic development laws and capital market advancements. Its core challenges and focal points evolve alongside societal progress [70]. From a micro perspective, principal-agent problems can occur between shareholders and management. It occurs when an agent exploits information asymmetry and conflicts of interest to harm the principal's interests. The principal, being at an informational disadvantage, is unable to fully monitor the agent [71]. Enterprise digital transformation alters the traditional principal-agent conflict between shareholders and management. This transformation inevitably reshapes management behavior.

On one hand, according to signaling theory, the capital market faces an issue of information asymmetry [14]. Enterprises undergoing digital transformation actively disclose information. And they can turn it into a tool for concept speculation and market value management among listed companies [72]. Nearly 70% of investors in the Chinese stock market are individuals. These individual investors often behave irrationally. Their irrational behaviors create a pronounced speculative atmosphere [73]. Individual investors, when processing information, are susceptible to the information cocoon effect and the ostrich effect [74,75]. And the heterogeneity of investor sentiment may make them respond differently. Investors may have expectations for future returns. When these expectations exceed their concerns about transformation risks, investor optimism increases. This increased optimism can lead to overconfidence. For instance, investors with anchoring bias might overlook crucial information. This information is needed for future considerations. Investors with representativeness bias may focus only on information they believe is representative. As a result, such biases can lead to impulsive decision-making. Conversely, if concerns about transformation risks surpass the anticipated returns, similar biases may result. Despite their preference for such trending topics, many fail to accurately discern and interpret overly positive tones. This behavior indirectly encourages managerial opportunism.

On the other hand, digital transformation is a long-term strategic plan. Digital transformation has inherent uncertainty. This uncertainty leads to increased business risks. Corporate risk information disclosure can help in predicting actual risks and eliciting market reactions [76,77]. Therefore, it becomes an important basis for investors' rational decision-making. Innovation is a fundamental aspect of digital transformation. Therefore, risk disclosures associated with digital transformation frequently include details on product development and technological innovation. Enhancing the redundancy of risk disclosure can increase firms' information transparency and reduce firms' agency costs [78]. And it can also positively affect firms in terms of investment efficiency [79] and IPO suppression [80]. But this information might become valuable incremental information utilizable by industry competitors

[81]. Meanwhile, the more risks disclosed, the stronger the market's risk perception becomes. This leads investors to be more cautious and adjust company valuations based on market risk disclosures [82]. Consequently, companies may opt to limit the disclosure of risk information [75,83,84]. Attribute substitution theory provides a suggestion [85]. It suggests that individuals use more readily available heuristic attributes. These heuristic attributes replace more complex and less accessible target attributes. That is, investors may lack direct access to corporate risk information. In this situation, they may use other information that is more conveniently obtained. This information helps them judge the company's future development. Textual tone is information that is easier to obtain compared to calculate financial indicators. Investors might think that a positive tone in the MD&A represents the management's confidence in the company's future development potential. And they may not be able to distinguish whether the tone overstates disclosures. Management can capitalize on this investor psychology by using a more positive tone in disclosures to draw investors' attention. Therefore, the first research hypothesis of this paper is proposed.

H1. All other factors being equal, corporate digital transformation significantly promotes to management's tone manipulation.

## 2.3. The moderating effect of media slant

With the development of the digital economy, obtaining media information from online platforms has become increasingly popular among investors. As an important participant in the capital market, financial media serve as information intermediaries and external supervisors. By collecting, processing, and disseminating information, financial media can enhance information transmission efficiency and reduce information asymmetry among market participants [86,87]. Furthermore, as external observers, they convey signals of corporate earnings sustainability to investors. In doing so, they protect investor interests. However, the media cannot always remain objective and neutral. For reasons such as attracting readers, pursuing commercial interests, or other motives, they may produce biased media sentiment. This bias can be unintentional or deliberate [88–90]. This sentiment can lead to reporting bias, affecting the accuracy and transparency of information. Finally, it weakens the media's external supervision function and encourages managerial opportunism. Moreover, the 'cover-up' effect of positive emotional information is notably significant [91].

On one hand, positive reporting conveys a non-factual optimistic sentiment. This sentiment aligns with the overly optimistic tone of management's information disclosure. The media plays a role in external supervision. When it endorses and views enterprise management positively, investors tend to exhibit heightened optimism. This occurs especially when enterprises undergoing digital transformation emit positive signals. On the other hand, media sentiment-induced reporting bias exacerbates the issue of information asymmetry. This sentiment effectively masks management's attempts at tone manipulation. By doing this, it diverts investor attention. As a result, a positive corporate image is fostered. Convinced investors tend to trust management's optimistic disclosures. As a result, investors neglect to investigate potential management errors. They also overlook the associated business risks during digital transformation. Therefore, this paper proposes its second research hypothesis.

H2. Media slant positively moderates the relationship between corporate digital transformation and management's tone manipulation. That is, as media's optimistic sentiment increases, digital transformation is more positively associated with tone manipulation.

Fig 1 presents the theoretical model, followed by details of its constituent elements.

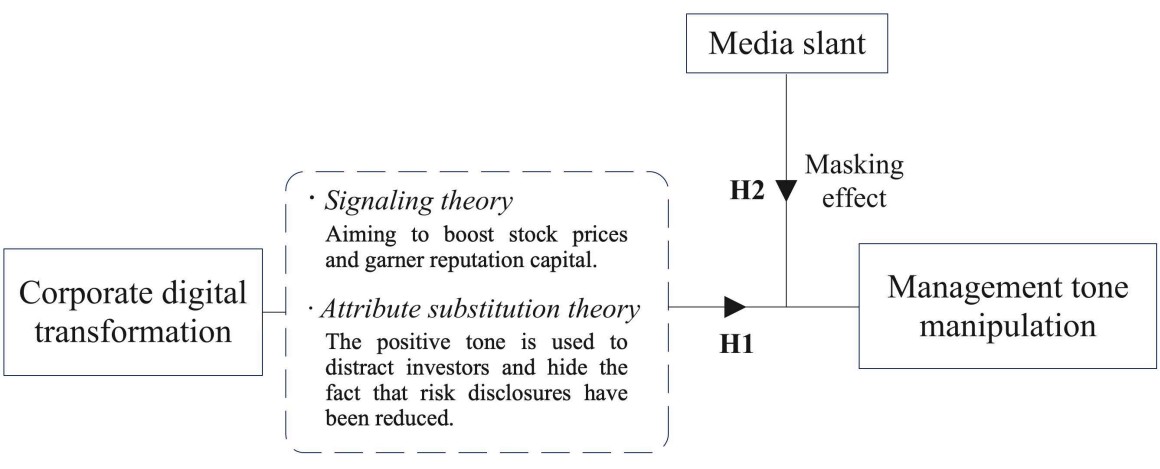

**Fig 1. Theoretical model.**

## 3. Data, variable constructions, and research design

### 3.1 Data and sample

The financial and capital market information environment data are obtained from the China Stock Market, Wind and CSMAR database. The digital transformation data are obtained from the annual reports of listed companies in China. MD&A text tone and media report data are obtained from CNRDS database.

In 2012, China revised *the No. 2 Standard on Information Disclosure Content and Format of Publicly Issued Securities Companies*. This revision affected the content and format of annual reports. It made the position, content, and form standards for MD&A disclosures more unified in the annual report. And thus made the contents of the MD&A chapters in the annual reports of listed companies more comparable. Therefore, A-share listed companies of China from 2012 to 2021 are selected to comprise the research sample. Considering the particularity of the industry characteristics, the financial industry, special treatment companies, and those with missing data on the main research variables are eliminated. The final research sample contains 19,202 firm-year observations. All continuous variables are Winsorised at the 1 percent level to avoid the influence of outliers.

### 3.2 Variables measurement

**3.2.1. Digital transformation of enterprises (Digital).** Referring to Qi et al. (2020) [92], this study's main regression analysis quantifies enterprise digital transformation by examining the proportion of digital assets in listed companies. Specifically, the notes to the financial statements in the annual report of the enterprise disclose the details of the intangible assets. We identify keywords related to digital technology, such as "software", "network", "client", "management system" and "smart platform". These items are classified as "digital intangible assets". We add up the number of digital technology intangibles for each company in each year. Then, we calculate the proportion of these intangibles relative to the company's total intangibles for the current year. This proportion is the proxy variable of enterprise digital transformation.

Additionally, we refer to Wu et al. (2021) [93]. We use the frequency of digital transformation keywords in the annual reports of listed companies. This frequency serves as a proxy measure of enterprise digital transformation during robustness testing. Specifically, through text analysis, we identify feature words from five categories: artificial intelligence, blockchain,

cloud computing, big data, and digital technology applications (Details in Appendix 1). Using Python 3.7.0, we extract keywords from each company's annual report and count the frequency of these keyword occurrences per company each year. Finally, we calculate the logarithm of the frequency plus one, which serves as a proxy indicator for the enterprise digital transformation.

**3.2.2. Tone manipulation.** *Calculate MD&A net positive intonation (Tone)*: With the development of text analysis technology, scholars built a lexicon with positive and negative emotions in the process of studying intonation. According to the frequency with which these vocabularies appear, a "net positive tone" is constructed to quantify the emotional tendencies of management [29,94]. The Harvard Dictionary's definition of negative words is not completely suitable for the tone analysis requirements in financial reports. To address this gap, Loughran and McDonald (2011) [94] developed a list of negative words suitable for analyzing financial report texts. Therefore, this paper relies on the vocabulary list developed by these two scholars. The study carries out translation through Youdao Dictionary and Jinshan Ciba, as well as retains the Chinese words which most related to the emotion expressed by these English words. And the final list of words included 1076 positive words and 2080 negative words.

There are two methods to calculate net positive intonation. For one thing, Davis and Tama-Sweet (2012) [35], Huang et al. (2014) [29] defined Tone as the number of positive words minus the number of negative words, and then divided by the total number of words in the annual report. For another, Henry (2008) [25], Price et al. (2012) [95] defined Tone as the number of positive words minus the number of negative words, and then divided by the sum of them. This research chooses the second one to measure the MD&A net positive intonation. A larger Tone value indicates a more positive sentiment in the text.

*MD&A tone manipulation (Abtone)*: To isolate unusually positive tones, we follow the methodology of Li (2010) [32], Huang et al. (2014) [29]. First, we select the control variables related to Tone to construct the model. Then, we use the residual method to divide Tone into two parts. One part is the normal tone. It reflects the company's fundamentals. The other part is the Abtone. It represents an unusually positive tone that deviates from the company's actual performance level. That is, the residual term from the regression captures the aspect of the text tone that surpasses the company's fundamental realities. It reflects the exaggerated emotional tendencies in management's expressions. This is used as a proxy for management's tone manipulation. The regression is:

$$
\begin{aligned}
\text{Tone}_{i,t} = \ &\beta_0 + \beta_1 Earn_{i,t} + \beta_2 Ret_{i,t} + \beta_3 Size_{i,t} + \beta_4 Btm_{i,t} + \beta_5 Chg\_Earn_{i,t} + \beta_6 Std\_Ret_{i,t} \\
&+ \beta_7 Busseg_{i,t} + \beta_8 Age_{i,t} + \beta_9 Loss_{i,t} + \beta_{10} Roa_{i,t} + \beta_9 Std\_Roa_{i,t} \\
&+ \varepsilon_{i,t}
\end{aligned} \tag{1}
$$

In the model, to control for financial performance, we include the following variables: Earn, which represents earnings after deducting extraordinary items scaled by lagged total assets; Ret, monthly stock return, which cash dividend reinvestment was taken into account; Chg_Earn, which represents volatility of performance, that is, the ratio of the change in net profit after deducting non-recurring gains and losses to total assets at fiscal year-begin; Loss, an indicator variable that is set to 1 if Earn has a negative value and 0 otherwise and Roa, which represents net profit by total assets.

We also include Size, which represents the natural logarithm of market value of equity at fiscal year-end; Std_Ret which represents standard deviation of monthly stock returns over the fiscal year; Std_Roa which represents standard deviation of Roa over five years in the [t − 2, t + 2]; Btm, which represents book-to-market ratio measured at fiscal year-end, to control

for growth opportunity; Busseg, an indicator variable equal to log (1 + number of business segments), or 1 if item is missing from Wind and CSMAR database; Age, an indicator variable equal to log (1 + age from the first year the firm has been listed).

Table 1 reports the regression results of Model (1). We find that Tone is more positive when the firm is profitable, growing, big, and has more business segments, fewer volatile stock returns and net profit, and shorter listing age. Abtone is the residual of Regression (1).

**3.2.3. Media slant (Slant).** Referring to the study of Li et al. (2022) [96], this research counts the number of positive and negative articles of each company, and then uses the following formula to measure media slant (Slant). The larger the Slant, the more optimistic the media slant.

$$Slant_{i,t} = \frac{PositiveCoverage_{i,t} - NegativeCoverage_{i,t}}{1 + TotalCoverage_{i,t}}$$

Where $PositiveCoverage_{i,t}$ and $NegativeCoverage_{i,t}$ represent the number of positive and negative articles on company i in year t. $TotalCoverage_{i,t}$ represents the total number of reports. If $Slant_{i,t}$ is greater than zero, it indicates media slant.

## 3.3. Research design

To test our hypothesis 1, we estimate the regression model (2) and investigate the effect of digital transformation of enterprises on MD&A tone manipulation.

$$\begin{aligned} Abtone_{i,t} = {} & \beta_0 + \beta_1 Digital_{i,t} + \beta_2 Size_{i,t} + \beta_3 Lev_{i,t} + \beta_4 Roa_{i,t} + \beta_5 Indep_{i,t} \\ & + \beta_6 Board_{i,t} + \beta_7 R\_Noa_{i,t} + \beta_8 Top1_{i,t} + \beta_9 BM_{i,t} + \beta_{10} Inst_{i,t} \\ & + \beta_{11} Dual_{i,t} + \beta_{12} Soe_{i,t} + \beta_{13} HHI_{i,t} + \beta_{14} Analyst_{i,t} + \beta_{15} Salary_{i,t} \\ & + \beta_{16} Std\_Ret_{i,t} + \varepsilon_{i,t} \end{aligned} \quad (2)$$

where, Abtone is the abnormal component of Tone, measured as the residual term from the normal MD&A tone estimation model, Model (1). In the model, our hypothesis translates into a positive coefficient on Digital.

To ensure that our results are not affected by other factors that are correlated with corporate disclosure practices, we include various control variables that are used in prior studies

**Table 1. Expected Tone Model.**

| Variables | Tone | |
|---|---|---|
| | Coefficient | t-stat |
| Earn | 0.4301*** | 30.823 |
| Ret | 0.0061*** | 3.4362 |
| Size | 0.0017** | 2.3567 |
| Btm | −0.0274*** | −6.9834 |
| Chg_earn | −0.0552*** | −3.3473 |
| sd_Ret | −0.0726*** | −5.2561 |
| Busseg | 0.0191*** | 7.0524 |
| Age | −0.0752*** | −32.9168 |
| Loss | −0.0091*** | −4.6802 |
| _cons | 0.5532*** | −36.2582 |
| N | 25955 | |
| Adj.R$^2$ | 0.1089 | |

[32,64]. For instance, we control for the reporting company's financial characteristics such as firm size (Size), operating performance (Roa), financial risk (Lev), growth potential (BTM), and operating volatility (R_Noa and Std_Ret). We include the number of analysts following (Analyst) to control for the firm's information environment, and the number of directors (Board), the proportion of outside directors (Indep), share proportion of the largest share-holder (Top1), CEO duality (Dual) and lnpay (Salary) to control for board characteristics. Besides, we also control for the shareholding ratio of institutional investors (Inst), the nature of property rights (Soe) and the degree of industry competition (HHI). Lastly, we add industry and year dummy variables to control for their fixed effects.

To test our hypothesis 2, based on Model (2), media slant (Slant) and the interaction between digital transformation and media slant (Digital×Slant) are added to estimate the regression model (3). If H2 is true, the coefficient of Digital×Slant in Model (3) should be significantly positive.

$$
\begin{aligned}
Abtone_{i,t} &= \beta_0 + \beta_1 Digital_{i,t} + \beta_2 Slant_{i,t} + \beta_3 Digital \times Slant_{i,t} + \beta_4 Size_{i,t} + \beta_5 Lev_{i,t} \\
&+ \beta_6 Roa_{i,t} + \beta_7 Indep_{i,t} + \beta_8 Board_{i,t} + \beta_9 R\_Noa_{i,t} + \beta_{10} Top1_{i,t} + \beta_{11} BM_{i,t} \\
&+ \beta_{12} Inst_{i,t} + \beta_{13} Dual_{i,t} + \beta_{14} Soe_{i,t} + \beta_{15} HHI_{i,t} + \beta_{16} Analyst_{i,t} + \beta_{17} Salary_{i,t} \\
&+ \beta_{18} Std\_Ret_{i,t} \\
&+ \varepsilon_{i,t}
\end{aligned} \tag{3}
$$

## 4. Empirical results and analysis

### 4.1 Descriptive statistics

Table 2, Panel A, presents the descriptive statistics for MD&A tone manipulation, digital transformation and other variables included in the main analyses. Consistent with the findings of Huang et al. (2014) [29], we observe that the mean and median values of Tone are optimistic (0.391 and 0.398). This implies that disclosure tone in the MD&A section of Chinese listed firms is generally relatively optimistic. Furthermore, the mean (median) value of Abtone is 0.0128 (0.0191) and minimum(maximum) value is −0.274 (0.242), which means that Abtone has a large fluctuation in the sample enterprise. This fluctuation provides a certain basis for the relevant research in this paper. For the variable of digital transformation (Digital), the mean value is 0.0942. This shows that the average proportion of digital assets in intangible assets of listed companies is 9.42%. There is still a big gap compared to China's digital economy, which accounts for more than 40% of GDP. The median(maximum) value of Digital is 0.0136 (1), which means that there are large differences in digital transformation among the sample enterprise. All those provide favorable conditions for this research.

Descriptive statistics on sub-samples are also conducted to test the impact of digital transformation on managerial tone manipulation. Table 2, Panel B, demonstrates a tendency in MD&A. This tendency occurs regardless of whether the enterprise implements a digital transformation strategy. MD&A tends to evaluate and analyze the company's past operating conditions in an optimistic tone. It also predicts and judges the future development trend optimistically in the annual report. Not only that, compared with non-digital transformation enterprises, the tone of MD&A of digital transformation enterprises is extremely positive. This initially shows that digital transformation has a promoting effect on the strategic disclosure behavior of the management, which is basically in line with the expectation of H1.

**Table 2. Descriptive statistics of variables.**

| variables | Panel A | | | | | Panel B | | |
|---|---|---|---|---|---|---|---|---|
| | Number of observations (N = 19202) | | | | | Digital transformation | Non-digital transformation | Mean difference t-test |
| | | | | | | N = 12416 | N = 6786 | |
| | Mean | Std.dev. | Minimum | Median | Maximum | Mean | Mean | |
| Tone | 0.391 | 0.118 | 0.0595 | 0.398 | 0.637 | 0.402 | 0.371 | −0.0315*** |
| Abtone | 0.012 | 0.111 | −0.274 | 0.019 | 0.242 | 0.024 | −0.007 | −0.0305*** |
| Digital | 0.094 | 0.216 | 0 | 0.014 | 1 | 0.118 | 0.0505 | −0.0676*** |
| Size | 22.25 | 1.271 | 20.06 | 22.07 | 26.34 | 22.32 | 22.14 | −0.1850*** |
| Lev | 0.422 | 0.203 | 0.057 | 0.414 | 0.873 | 0.418 | 0.43 | 0.0125*** |
| Roa | 0.041 | 0.061 | −0.225 | 0.038 | 0.214 | 0.042 | 0.038 | −0.0045*** |
| Indep | 0.375 | 0.053 | 0.333 | 0.357 | 0.571 | 0.377 | 0.371 | −0.0061*** |
| Board | 2.13 | 0.193 | 1.609 | 2.197 | 2.639 | 2.122 | 2.145 | 0.0229*** |
| Top1 | 0.344 | 0.145 | 0.092 | 0.323 | 0.736 | 0.34 | 0.351 | 0.0109*** |
| R_Noa | 0.515 | 0.5 | 0 | 1 | 1 | 0.48 | 0.581 | 0.1008*** |
| Btm | 1.069 | 1.183 | 0.0955 | 0.681 | 7.359 | 1.043 | 1.117 | 0.0737*** |
| Inst | 0.39 | 0.235 | 0.001 | 0.395 | 0.884 | 0.393 | 0.383 | −0.0101*** |
| Dual | 0.281 | 0.449 | 0 | 0 | 1 | 0.301 | 0.245 | −0.0560*** |
| Soe | 0.342 | 0.474 | 0 | 0 | 1 | 0.313 | 0.395 | 0.0817*** |
| HHI | 0.194 | 0.166 | 0.041 | 0.14 | 1 | 0.199 | 0.185 | −0.0139*** |
| Analyst | 1.448 | 1.187 | 0 | 1.386 | 3.807 | 1.535 | 1.29 | −0.2441*** |
| Salary | 15.6 | 0.691 | 13.86 | 15.57 | 17.61 | 15.69 | 15.43 | −0.2530*** |
| Std_Ret | 0.126 | 0.059 | 0.043 | 0.113 | 0.369 | 0.127 | 0.125 | −0.0027*** |

## 4.2 Results of the empirical analysis

**4.2.1 Digital transformation and Abtone.** Table 3 presents the results of the baseline OLS regression using Model (3) after controlling for other potential determinants. In column (1), the explained variable Abtone is replaced with Tone firstly, we find a significant and positive coefficient on Digital suggesting that the digital transformation of enterprises can significantly promote the optimism degree of MD&A tone, which initially verifies the expectation of this research. Column (2) shows the regression result of Digital and Abtone (H1). Consistent with our expectation, the results show that the coefficient of Digital is significantly positive, indicating that digital transformation of enterprise is positively associated with upward tone management of MD&A in the annual reports. The estimated coefficient (coeff. = 0.0132, p < 0.01) implies that, on average, a one-standard-deviation increase in Digital is associated with an increase of 2.57% (= 0.0132 × 0.216/0.111) of a standard deviation in Abtone.

In terms of the control variables, Lev, Roa, Indep, Soe, Analyst and Salary are significantly positively associated with Abtone, while Btm, Inst and HHI are significantly and negatively related to Abtone. Specifically: (1) A higher debt ratio suggests that the company faces relatively high financial risks and requires additional funds to ease debt pressures. (2) A higher return on total assets signifies robust current profitability, making managers more confident to invest in risky projects [97], which consequently increases the need for additional funds. (3) Firms with more preoccupied independent directors have declining firm valuation and operating performance and exhibit weaker accounting quality [98]. (4) Executives in state-owned enterprises hold dual roles as both "professional managers" and "government officials". They

**Table 3. Digital transformation and abnormal MD&A tone.**

| Variable | (1) Tone | | (2) Abtone | |
|---|---|---|---|---|
| | Coefficient | T value | coefficient | T value |
| Digital | 0.0145*** | 3.55 | 0.0132*** | 3.30 |
| Size | −0.0007 | −0.54 | −0.0009 | −0.79 |
| Lev | 0.0197*** | 3.52 | 0.0251*** | 4.52 |
| Roa | 0.4103*** | 26.14 | 0.0397** | 2.50 |
| Indep | 0.0598*** | 3.31 | 0.0541*** | 3.02 |
| Board | 0.0043 | 0.83 | 0.0077 | 1.50 |
| R_Noa | 0.0011 | 0.66 | 0.0005 | 0.34 |
| Top1 | 0.0200*** | 3.37 | 0.0003 | 0.05 |
| Btm | −0.0089*** | −8.12 | −0.0053*** | −4.81 |
| Inst | −0.0127*** | −3.07 | −0.0126*** | −3.04 |
| Soe | 0.0069*** | 3.37 | 0.0160*** | 7.73 |
| Dual | 0.0040** | 2.27 | 0.0014 | 0.79 |
| HHI | −0.0374*** | −6.46 | −0.0294*** | −5.09 |
| Analyst | 0.0131*** | 15.49 | 0.0073*** | 8.63 |
| Salary | 0.0071*** | 4.78 | 0.0082*** | 5.57 |
| Sd_Ret | −0.0470*** | −2.92 | −0.0162 | −1.01 |
| Year & Firm FE | Yes | | | |
| _cons | 0.2546*** | 8.58 | −0.1383*** | −4.66 |
| N | 19202 | | | |
| R-squared | 0.21 | | 0.10 | |
| F | 121.95 | | 49.38 | |

Note: *, **, *** indicate statistical significance levels of 0.10, 0.05, and 0.01, respectively.

benefit not only from salary incentives but also from opportunities for political advancement. Implicit incentives for political advancement influence SOE executives. These incentives lead executives to focus on building and preserving personal and corporate images. As a result, their motivation for tone manipulation in annual reports is intensified [99]. (5) With higher compensation, management develops a strong opportunistic drive to defend their existing high wages and justify future raises. This drive to rationalize their compensation affects how the company discloses information [100,101]. In these situations, managers may manipulate tone, and this aligns with findings from prior research. Notably, both Analyst and HHI are proxies for external oversight, yet the regression coefficient for Analyst is positive. This finding aligns with [102,103]. This is because companies, to ease the increased pressure from more analysts' oversight, tend to disclose more exaggerated tones.

Other, (1) A high market-to-book ratio suggests significant investment value and growth potential, naturally drawing numerous potential investors [43]. (2) Institutional investors, in contrast to individual investors, possess higher expertise and market sensitivity, allowing them to more readily detect opportunistic management behaviors [104]. (3) Industry competition acts as an external governance mechanism, restricting opportunistic managerial behaviors [50]. In these scenarios, managers tend to reduce tone manipulation, aligning with findings from existing studies.

**4.2.2 Effect of media slant.** The estimated regression coefficients of Model (3) are collected in Table 4, which show that the positive relationship between Digital and Abtone remains significant at the 1% level. Moreover, the coefficients of the interaction between

**Table 4. Effect of media slant.**

| Variable | Abtone | |
|---|---|---|
| | Coefficient | T value |
| Digital | 0.0151*** | 3.83 |
| Slant | 0.0581*** | 16.77 |
| Digital×Slant | 0.0323** | 2.24 |
| Size | −0.0004 | −0.35 |
| Lev | 0.0216*** | 3.94 |
| Roa | −0.0075 | −0.47 |
| Indep | 0.0586*** | 3.29 |
| Board | 0.0069 | 1.35 |
| R_Noa | 0.0010 | 0.63 |
| Top1 | −0.0000 | −0.00 |
| Btm | −0.0053*** | −4.88 |
| Inst | −0.0109*** | −2.63 |
| Soe | 0.0145*** | 7.04 |
| Dual | 0.0018 | 1.02 |
| HHI | −0.0250*** | −4.39 |
| Analyst | 0.0043*** | 4.96 |
| Salary | 0.0085*** | 5.84 |
| Sd_Ret | −0.0094 | −0.59 |
| Year & Firm FE | Yes | |
| _cons | −0.1550*** | −5.27 |
| N | 19194 | |
| R-squared | 0.1092 | |
| F | 55.29 | |

Digital and Slant are positive and significant at the 5% level. The results are consistent with our expectation. They show a positive relation between the digital transformation of enterprises and abnormal MD&A tone. This positive relation is more prominent when the media slant is more optimistic.

## 4.3 Robustness tests

**4.3.1 Endogeneity tests using the treatment effect approach.** Endogeneity problem may arise from unconscious variable omitting [105,106], which results in the correlation between the main explanatory variable and error term, i.e., Cov (HY, ε) ≠ 0. Thus, a Heckman two-stage procedure is adopted to control for the endogeneity issue and re-test H1. In the first step, a probit model is estimated. The dependent variable (digital) equals 1 if a firm applies digital transformation. This means the annual report contains relevant vocabulary. It equals 0 otherwise. Partial control variables in the main tests are included in the first stage estimation. At the same time, other factors affecting digital transformation are controlled. These factors include TobinQ, FirmAge, Cashflow, and Opinion. The Inverse Mill's Ratio (IMR) for each firm-year observation is calculated in the first stage. It is included in the second stage of the Heckman estimation. The Inverse Mill's Ratio (IMR) for each firm-year observation is calculated in the first stage and included in the second stage of the Heckman estimation.

Panel A of Table 5 reports the first-stage Probit regression and the second stage regression results, Following Lennox et al. (2012) [107], we include the Inverse Mill's Ratio in the second

stage to control for the endogeneity issue. As shown in column (2) of Panel A, the coefficient on Digital is significantly positive (coeff. = 0.0136, p < 0.01), lending additional support to H1. Furthermore, it is worth noting that the IMR has significant coefficients in column (2), suggesting the necessity of using the Heckman two-stage procedure to mitigate the potential impacts of omitting variables.

**4.3.2 Endogeneity tests using the propensity score matching approach.** In this section, we employ the propensity score matching (PSM) approach. This approach helps alleviate the endogenous problem caused by sample selection bias. Some basic characteristics of enterprises may affect both enterprise digital transformation and management's strategic information disclosure, influencing the relationship between the two. The samples that underwent digital transformation during the sample period are used as the treatment group to make the sample more comparable. The control group includes companies that did not undergo digital transformation during the sample period. Specifically, the Digital variable is one-to-one and non-repeated matched by Size, Lev, Roa, Dual, TobinQ, FirmAge, Top1, Opinion, Mfee and Inst, while simultaneously controlling the fixed effect of year and industry.

The standardized deviation of covariates after matching is much smaller than that before matching, and the maximum deviation is 1.9%, which is also much smaller than the empirical value of 10%. And there is no significant difference between the treatment group and the control group, so the balance hypothesis test is passed.

Panel B of Table 5 reports the results of Hypotheses 1 using the PSM sample and shows that the coefficient of Digital and Abtone remains significantly positive (coeff. = 0.0193, p < 0.05). It suggests our main results are robust to using the PSM method, verifying H1 again.

**4.3.3 Other robustness tests.** This paper further adopts the following tests to improve the robustness of the conclusions: (1) Alternative Definition of Digital Transformation of

**Table 5. The endogeneity treatment.**

| Panel A: Heckman two-stage treatment effect regression | | | | |
|---|---|---|---|---|
| | The First Stage | | The Second Stage | |
| Variable | digital | | Abtone | |
| | (1) | | (2) | |
| | Coefficient | Z value | Coefficient | T value |
| IMR | – | – | 0.0465*** | 4.93 |
| Digital | – | – | 0.0136*** | 3.38 |
| _cons | −3.5714*** | −12.84 | −0.2500*** | −6.65 |
| Year & Firm FE | Yes | | | |
| N | 19,202 | | | |
| R-squared | – | | 0.10 | |
| F | – | | 48.29 | |
| **Panel B: Regression analysis based on PSM sample** | | | | |
| Variable | Abtone | | | |
| | Coefficient | | T value | |
| Digital | 0.0193** | | 2.47 | |
| _cons | −0.0350 | | −0.63 | |
| Controls | Yes | | | |
| Year & Firm FE | Yes | | | |
| N | 10252 | | | |
| Adj.R2 | 0.13 | | | |
| F | 26.00 | | | |

Enterprise (digital). Following Wu et al. (2021) [93], we define digital transformation as the logarithm of the total frequency of digital transformation keywords. (2) Alternative Definition of Abnormal MD&A Tone (abtone). We construct a dummy variable for Abnormal MD&A Tone, equaling 1 if the regression residual of Model (1) is positive and 0 otherwise. (3) Alternative Time Window Based. Digital transformation is a long-term strategic change. The information disclosure in annual reports has a certain lag. Therefore, this paper extends the time window for investigating the impact of enterprise digital transformation on management's strategic information disclosure. The digital transformation index is lagged by one period. (4) Alternative Model of Regression H1. To avoid the possible influence of different regression methods on the research results, we use Tobit Model to replace the mixed OLS regression for robustness test. And we control the year fixed effect and the clustering standard error at the city level. (5) Excluding Some Sample Companies. The high level of digital infrastructure of high-tech enterprises, may affect the universality of the conclusion. While Growth Enterprises Market (GEM or second board) and Sci-Tech innovation board (STAR Market) are mostly high-tech listed companies, which are closely related to digitalization and informatization. Therefore, after excluding the samples of GEM and STAR Market listed companies in A-share market, Model (2) is used to regress the samples. Columns (1)-(5) in Table 6 report the test results in turn, and the results show that the conclusions of this paper are robust.

## 5. Further analyses

Further analyses involve conducting heterogeneity tests from the perspectives of financial flexibility and risk information disclosure. These explore the motivations behind tone manipulation by management in digitally transforming enterprises. These findings further validate the internal logic of the hypothesis. Additionally, the study examines how digital transformation, through management overconfidence, influences tone manipulation.

### 5.1 Financial flexibility

Financial flexibility enables a company to mobilize internal financial resources, preventing and mitigating future uncertainties [108]. According to pecking order theory, companies prioritize internal financing [109]. In uncertain environments, financial flexibility's low-cost advantage acts as a buffer. It enables companies to seize investment opportunities during crises. It helps companies withstand external risks. It also allows companies to allocate resources to promising projects and pursue long-term development [110]. Additionally, the level of financial flexibility reflects the capabilities and enthusiasm of management [111]. The capital chain might break

Table 6. Robustness Tests.

| Variable | (1) | (2) | (3) | (4) | (5) |
|---|---|---|---|---|---|
|  | Abtone | Abtone | Abtone | Abtone | Abtone |
| digital | 0.217*** | 0.032*** | 0.012*** | 0.011** | 0.019*** |
|  | 2.59 | 18.41 | 2.63 | 2.32 | 3.92 |
| _cons | −1.882*** | −0.113*** | −0.121*** | 0.009*** | −0.214*** |
|  | −3.27 | −3.85 | −3.51 | 50.42 | −6.50 |
| Year & Firm FE | Yes |  |  |  |  |
| N | 19202 | 19202 | 14804 | 19202 | 15358 |
| R-squared | – | 0.109 | 0.086 | – | 0.100 |
| F | – | 59.014 | 35.118 | 75.547 | 43.976 |

and funding gaps may persist. If these issues occur, digital transformation efforts may fail. This failure negatively impacts the company. It also negatively impacts its management. Facing financial pressure, companies may urgently seek to enhance short-term benefits to maintain their image and secure financing. This urgency can lead to managerial short-sightedness, with information manipulation being a key strategic tool. Management might manipulate the tone of disclosed information to attract investors. They aim to showcase a promising future and efficient management. This can obscure any unfavorable information. It also masks the real business situation. According to Arslan et al. (2014) [112], this paper adopts the dual-indicator approach to measure financial flexibility. And the formulas are as follows.

$$Financial\ Flexibility = Cash\ Flexibility\ +\ Liability\ Flexibility$$

$$Cash\ Flexibility = Corporate\ cash\ ratio\ -\ industry\ average\ cash\ ratio$$

$$Liability\ Flexibility = max(0,\ industry\ average\ debt\ ratio$$

$$-\ corporate\ debt\ ratio)$$

The full sample enterprises are grouped according to financial flexibility (Ff). When the sample firms' Ff is higher than the industry median for the year, they are classified as strong financial flexibility firms, otherwise they are classified as weak financial flexibility firms.

The regression results are shown in Panel A of Table 7. The Digital regression coefficients are all significantly positive at the 1% level in the financially fewer flexible firms, while are positive but not significant in the financially more flexible firms. The two sets of regression results passed the test of between-group differences. This fully validates the conjecture and deduction of previous articles. The effect of corporate digital transformation in promoting strategic management disclosure is more pronounced in firms with less financial flexibility.

## 5.2  Risk disclosure

Risk information disclosure not only affects investors' risk perception, but also impacts their decision-making [113,114]. Investors can adjust their investment decisions. They revise their expectations for the company's future development based on risk information disclosure. Therefore, companies may reduce risk information disclosure. Management may enhance the text's optimistic tone. By doing this, they can guide investors towards favorable decisions. An abnormally optimistic tone may not reflect the company's actual operating conditions. Non-fully rational investors may be swayed by herd mentality. These investors might not invest the necessary time and energy. They may fail to verify the authenticity of the information. This leads to indiscriminate praise of the "digital" concept and susceptibility to overreaction and misjudgment of overly positive tones [115]. Consequently, investor blindness also offers managers an opportunity to manipulate tone.

The full sample enterprises are grouped according to risk disclosure (Risk), which is measured by the text analysis method. Sample firms are categorized based on their risk. If a firm's risk is higher than the industry median for the year, it is categorized as a higher risk disclosure level firm. If a firm's risk is not higher than the median, it is categorized as a lower risk disclosure level firm. The regression results are shown in Panel B of Table 7. In firms with lower levels of risk disclosure, the Digital regression coefficient is significantly positive at the 1% level. In firms with higher levels of risk disclosure, the Digital regression coefficient is positive but insignificant. And the coefficient is smaller than that of firms with lower levels of risk disclosure. This is consistent with our expectation that firms with lower levels of risk disclosure are more likely to

**Table 7.  Heterogeneity Tests.**

| Panel A: Corporate financial flexibility | | | |
|---|---|---|---|
| **Variable** | **Abtone** | | **Intergroup difference test** |
| | **More financial flexibility** | **Fewer financial flexibility** | |
| Digital | 0.0043 | 0.0237*** | 0.019*** |
| | 0.80 | 3.86 | |
| _cons | −0.1621*** | −0.1112** | 0.051 |
| | −3.87 | −2.62 | |
| Year & Firm FE | Yes | | |
| N | 9660 | 9542 | – |
| R-squared | 0.1057 | 0.0971 | – |
| F | 28.10 | 24.91 | |
| Panel B: Enterprise risk information disclosure | | | |
| Variable | Abtone | | Intergroup difference test |
| | higher risk disclosure | lower risk disclosure | |
| Digital | 0.0062 | 0.0155*** | 0.009 |
| | 1.13 | 2.90 | |
| _cons | −0.2021*** | −0.0595 | 0.143*** |
| | −5.01 | −1.51 | |
| Year & Firm FE | YES | | |
| N | 9,602 | 9,600 | – |
| R-squared | 0.0889 | 0.1271 | – |
| F | 22.31 | 34.76 | |

engage in MD&A text tone manipulation. However, the two regressions do not pass the intergroup difference test. It maybe because digital transformation also enhances firms' risk-taking levels and higher risk-taking levels may also influence management disclosure decisions.

## 5.3  Mechanism analysis

Research on overconfidence began in psychology. This research has gradually expanded to other fields. It now includes economics, management, and more [116,117]. Social psychology views overconfidence as an individual's overly optimistic, irrational judgment about oneself or situations. It is primarily manifested in two ways [118]. The first way is through optimistic psychological expectations. This means overestimating the likelihood of favorable outcomes. The second way is the overestimation of one's abilities. This involves an excessive confidence in the effectiveness and reliability of one's judgment. This confidence exists without considering personal bias. Overconfidence is a cognitive bias in decision-making. It is prevalent among corporate managers [119]. This overconfidence epitomizes managers' irrational behavior. According to the overconfidence hypothesis and stewardship theory, cognitive biases lead overconfident managers to maintain an optimistic view of the market and the company's prospects [120]. They often overestimate their management skills, underestimate potential risks, and struggle to respond rationally to negative management feedback [121].

Previous studies have shown that past experiences and external environments are the two main sources of management's overconfidence. On the one hand, corporate management serves as decision-makers. They possess significant control over resources. They have extensive professional knowledge and an experience of success. They enjoy social reputation. These factors can lead managers to overestimate their decision-making capabilities [122].

Additionally, digital transformation represents a strategic overhaul that leverages a company's full capabilities and potential [1]. Digital transformation involves decision-making. This process reinforces managers' confidence in their decision-making abilities. This reinforcement can potentially lead to overconfidence. On the other hand, favorable evaluations and positive feedback from the external environment will increase the degree of management's overconfidence [123]. Under the digital economy's influence, investors have high expectations for the novel concept of digitalization. Consequently, this optimism towards digital transformation makes it an effective strategy. It helps companies garner market attention and improve market feedback [124]. The positive external reaction further amplifies the degree of management's overconfidence.

At the same time, an individual's cognitive biases not only produce overconfidence but also affect individual behavior. For example, management's overconfidence can affect the accounting conservatism of its information disclosure [125]. First, managers with overconfidence tend to have overly optimistic views on the company's future. This optimism motivates them to send positive signals to investors. Signal transmission theory suggests this behavior [116]. Secondly, the process of digital transformation in enterprises requires continuous and substantial capital investment [126]. The overly optimistic psychology of management may lead them to participate in some inefficient resource allocation projects. Information asymmetry theory provides a basis for management's actions [92]. Finally, management is driven by self-interest to build a high-reputation image. They might exaggerate the impact of digital transformation, leveraging optimistic tones to engage investors' emotions. They aim for benefits like position promotion and justifying on-the-job consumption [127].

This study draws upon existing research [128], employing the profit forecast levels of listed companies as indicators of managerial overconfidence. Specifically, it statistically analyzes the profit forecast information disclosed by sample companies between 2012 and 2021, including 1st quarterly, semi-annual, 3rd quarterly, and annual reports. Some profit forecast information reported by companies is essentially "pre-announcements". This information is disclosed after the relevant period has ended. This study excludes "pre-announcement" samples. In these cases, management already knows the actual profits before forecasting. This does not accurately reflect overconfidence. In order to examine the mechanism of the impact of digital transformation on the financial distress of enterprises, this paper uses model (2), model (4) and model (5) to test step by step:

$$
\begin{aligned}
\text{Confidence}_{i,t} &= \beta_0 + \beta_1 \text{Digital}_{i,t} + \beta_2 \text{Size}_{i,t} + \beta_3 \text{Lev}_{i,t} + \beta_4 \text{Roa}_{i,t} + \beta_5 \text{Indep}_{i,t} + \beta_6 \text{Board}_{i,t} \\
&\quad + \beta_7 \text{Noa}_{i,t} + \beta_8 \text{Top1}_{i,t} + \beta_9 \text{BM}_{i,t} + \beta_{10} \text{Inst}_{i,t} + \beta_{11} \text{Dual}_{i,t} + \beta_{12} \text{Soe}_{i,t} \\
&\quad + \beta_{13} \text{HHI}_{i,t} + \beta_{14} \text{Analyst}_{i,t} + \beta_{15} \text{Salar}_{i,t} + \beta_{16} \text{Std\_Ret}_{i,t} \\
&\quad + \varepsilon_{i,t}
\end{aligned} \tag{4}
$$

$$
\begin{aligned}
\text{Abtone}_{i,t} &= \beta_0 + \beta_1 \text{Digital}_{i,t} + \beta_2 \text{Confidence}_{i,t} + \beta_3 \text{Size}_{i,t} + \beta_4 \text{Lev}_{i,t} + \beta_5 \text{Roa}_{i,t} \\
&\quad + \beta_6 \text{Indep}_{i,t} + \beta_7 \text{Board}_{i,t} + \beta_8 \text{Noa}_{i,t} + \beta_9 \text{Top1}_{i,t} + \beta_{10} \text{BM}_{i,t} \\
&\quad + \beta_{11} \text{Inst}_{i,t} + \beta_{12} \text{Dual}_{i,t} + \beta_{13} \text{Soe}_{i,t} + \beta_{14} \text{HHI}_{i,t} + \beta_{15} \text{Analyst}_{i,t} \\
&\quad + \beta_{16} \text{Salary}_{i,t} + \beta_{17} \text{Std\_Ret}_{i,t} \\
&\quad + \varepsilon_{i,t}
\end{aligned} \tag{5}
$$

Table 8, column (1) reveals a positive correlation between Digital and Confidence (0.026, $p < 0.1$), suggesting digital transformation elevates management overconfidence. Furthermore, when including Digital and Confidence in the regression model, column (2) demonstrates that both Digital and Confidence significantly positively influence Abtone (0.013 and 0.016,

**Table 8. Mechanism analysis.**

| Variable | (1) | (2) |
|---|---|---|
| | Confidence | Abtone |
| Digital | 0.026* | 0.013*** |
| | 1.76 | 3.2 |
| Confidence | – | 0.016*** |
| | | 8.62 |
| _cons | 1.206*** | −0.158*** |
| | 11.26 | −5.32 |
| Year & Firm FE | Yes | |
| N | 19202 | 19202 |
| Adj.R² | 0.36 | 0.097 |
| F | 555.083 | 50.295 |

respectively, both $p < 0.01$). Consequently, management overconfidence serves as a partial mediator in this relationship. This suggests that corporate digital transformation not only directly influences management tone manipulation but also indirectly promotes it by fostering management overconfidence.

# 6. Conclusions and practical implications

## 6.1 Conclusions

Based on impression management, signal transmission, and attribute substitution theories, this study focuses on the "Management Discussion and Analysis" sections from the annual reports of Chinese A-share listed companies between 2012 and 2021. Examining the impact of digital transformation on management's tone manipulation behavior, the main research work and conclusions of this study are as follows:

(1) Groups were divided based on digital transformation status to examine differences in MD&A tone manipulation. Findings reveal that companies tend to portray their operations and prospects optimistically in MD&A reports. This is true whether they have implemented digital transformation strategies. Digitally transformed companies show higher and more pronounced optimism in their MD&A reports than non-transformed ones. Transformed companies are more motivated to manage MD&A tone for specific objectives compared to non-transformed companies.

(2) In the digital economy's exploration phase, corporate digital transformation notably encouraged management's tone manipulation. Specifically, management is motivated by signaling theory. They amplify successes of digital transformation through tone manipulation. They also obscure its challenges. Their goal is to elicit positive market reactions to their stock price.

(3) The more optimistic the media sentiment is, the greater its impact on how management manipulates tone during digital transformation. Media reports that lack objectivity and neutrality lead to biased sentiment. This bias undermines their roles in information mediation and external oversight. Positive reports are sometimes misaligned with the reality of digital transformation enterprises. These reports divert investors' focus from operational risks to optimistic views. They also mask overly optimistic text information. This optimism is often steered by management's tone manipulation.

(4) Companies with limited financial flexibility show a stronger correlation between digital transformation and tone manipulate. In uncertain markets and competitive environments, financial flexibility provides cost benefits. These benefits help ease the funding pressures faced by firms undergoing digital transformation. Firms with less financial flexibility undergoing digital transformation might encourage short-sighted managerial actions. These actions include using tone manipulation to blur negative information and emit positive signals for specific goals.

(5) Firms with less risk information disclosure show a stronger link between digital transformation and the manipulation of management's communication tone. Firms that reduce risk disclosure might use a more optimistic tone in their texts. This optimistic tone can influence investors towards decisions that benefit the firm.

(6) Managerial overconfidence partially mediates the relationship between digital transformation and tone manipulation. Implementing a digital transformation strategy can boost managerial overconfidence, subsequently impacting their behaviors. One manifestation of this is tone manipulation. The objective is to steer investors toward decisions that benefit management.

## 6.2  Practical implications

This paper has some implications for the government, regulators and investors:

(1) Policymakers can develop targeted policies. They can base these policies on successful experiences. These policies address enterprises' financial challenges. They also tackle key obstacles in digital transformation. This approach facilitates the transformation process and swiftly harnesses the competitive edge digitalization offers.

(2) Regulators should focus on the form and content of information disclosure, setting clear standards for substantive content. By evaluating disclosure quality from various angles, they can aid enterprises in reducing disclosure costs and enhancing proactive risk information disclosure without revealing proprietary details. Simultaneously, they can enhance oversight of listed companies' disclosures. Increasing the costs of strategic disclosures can also help. These actions can mitigate capital market information asymmetry and elevate disclosure quality.

(3) Investors should carefully assess the optimistic tone in MD&A by comparing financial data and textual information. This comparison helps gauge tone anomalies and avoid being misled by strategic disclosures. They should also critically evaluate media report objectivity by comparing different sources. Furthermore, investors should focus on negative and risk information. They should also enhance their financial literacy through academic reports and lectures to reduce the risk of being misled by strategic disclosures.

## 6.3  Limitations and prospects

This study also has several limitations, with future research opportunities in these main areas:

(1) Different digital transformation types focus on varied aspects. These types include process, service, and organizational transformations. They can potentially affect information disclosure differently. Incorporating these types into the research framework allows for a comprehensive impact analysis.

(2) The paper examines digital transformation's influence on MD&A text tone. Exploring whether "digital transformation" as an optimistic signal, is exploited in strategic disclosures warrants further research.

(3) Although this study maintains an adequate sample size, its findings are limited in generalizability as it focuses solely on Chinese A-share listed companies. Countries and markets differ significantly in their information disclosure regulations and stages of digital transformation. Future research should broaden its scope. It should thoroughly investigate the specific circumstances of various nations concerning information disclosure regulation and digital transformation.

**Declaration of generative AI and AI-assisted technologies in the writing process.** During the preparation of this work the authors used GPT-4 model from OpenAI via API in order to translate the manuscript from Chinese to English, improve readability and language. After using this service, the authors reviewed and edited the content as needed and take full responsibility for the content of the publication.

## Supporting information

**S1 File. Supporting Information-Appendix 1.**
(DOCX)

## Author contributions

**Conceptualization:** Lingyun Yang, Zhihong Zhang, Lulu Wang.

**Data curation:** Lingyun Yang.

**Formal analysis:** Lingyun Yang.

**Funding acquisition:** Zhihong Zhang.

**Investigation:** Lingyun Yang.

**Methodology:** Lingyun Yang, Lulu Wang.

**Project administration:** Zhihong Zhang.

**Software:** Lingyun Yang.

**Supervision:** Zhihong Zhang, Lulu Wang, Yikai Liang.

**Validation:** Zhihong Zhang.

**Writing – original draft:** Lingyun Yang.

**Writing – review & editing:** Lingyun Yang, Zhihong Zhang, Lulu Wang, Yikai Liang.

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
