## [Decision Letter · Decision Letter 0]

17 Oct 2024

PONE-D-24-36330

Encourage or inhibit: A Study on the Impact of Corporate Digital Transformation on Management's Tone Manipulation of Information Disclosure

PLOS ONE

Dear Dr. Zhang,

Thank you for submitting your manuscript to PLOS ONE. After careful consideration, we feel that it has merit but does not fully meet PLOS ONE’s publication criteria as it currently stands. Therefore, we invite you to submit a revised version of the manuscript that addresses the points raised during the review process.

We look forward to receiving your revised manuscript.

Kind regards,

Reza Rostamzadeh

Academic Editor

PLOS ONE

Journal Requirements:

2. Thank you for stating the following financial disclosure: “This work was supported by the Humanities and Social Sciences Project of the Ministry of Education of China (No.22YJA630118) .”

3. Please note that your Data Availability Statement is currently missing the repository name and/or the DOI/accession number of each dataset OR a direct link to access each database. If your manuscript is accepted for publication, you will be asked to provide these details on a very short timeline. We therefore suggest that you provide this information now, though we will not hold up the peer review process if you are unable.

Additional Editor Comments:

Dear Prof. Zhang

We have now completed the reviewing process of your article, and we are pleased to say that we consider it a good candidate for publication, once a number of revisions are made.

Please read the reviewers' recommendations listed below and revise your article in light of their comments.

Kind regards

Reza Rostamzadeh

Reviewers' comments:

Reviewer's Responses to Questions

**Comments to the Author**

1. Is the manuscript technically sound, and do the data support the conclusions?

Reviewer #1: Yes

Reviewer #2: Partly

Reviewer #3: Yes

2. Has the statistical analysis been performed appropriately and rigorously? 

Reviewer #1: Yes

Reviewer #2: Yes

Reviewer #3: Yes

3. Have the authors made all data underlying the findings in their manuscript fully available?

Reviewer #1: Yes

Reviewer #2: Yes

Reviewer #3: No

4. Is the manuscript presented in an intelligible fashion and written in standard English?

Reviewer #1: Yes

Reviewer #2: Yes

Reviewer #3: Yes

5. Review Comments to the Author

Reviewer #1: The manuscript provides a comprehensive analysis of the impact of corporate digital transformation on management's tone manipulation in information disclosure, offering valuable insights into how digitalization affects corporate communication strategies. Here are detailed comments to support the evaluation:

Areas for Improvement:

Clarity in Variable Definition: While the methodology is sound, some variable definitions could be clearer, particularly for readers who may not be familiar with the context of the Chinese market. It would be beneficial to provide more details on the calculation of variables such as 'Digital' and 'Abtone' in the methodology section.

Generalisability of Findings: The study focuses on Chinese A-share listed companies. Although the sample size is large, the findings may not be directly generalizable to other countries or markets with different regulatory environments or digital transformation stages. Discussing this limitation in more detail would strengthen the study's implications.

Recent Literature: While the literature review is comprehensive, incorporating more recent studies (post-2021) on digital transformation and tone manipulation could enhance the relevance and impact of the manuscript.

Discussion of Findings: Although the results are presented clearly, the discussion could benefit from a deeper analysis of why certain control variables, such as 'Lev,' 'Indep,' and 'Soe,' have significant effects on abnormal tone. This would provide more context and depth to the findings.

Ethical Considerations:

There appear to be no concerns regarding research ethics or potential conflicts of interest. The manuscript clearly acknowledges the use of AI-assisted technologies for translation and readability improvements, ensuring transparency.

Minor Issues:

Language and Grammar: There are a few minor grammatical and typographical errors. For example, ensure consistency in tenses and correct any punctuation issues in the manuscript.

Figure and Table Presentation: Some tables, such as Table 6, could benefit from clearer formatting or highlighting key findings to improve readability.

Overall Evaluation: The manuscript is technically sound, methodologically rigorous, and offers valuable contributions to the literature on digital transformation and information disclosure. It has the potential to significantly impact both academic research and practical applications. Addressing the minor issues and providing more detailed discussions on certain aspects will further enhance the quality and readability of the paper.

Reviewer #2: In quotes, you will find parts of the text copied and pasted here in order to give you a reference to my commentary inside the paper.

"1. Introduction"

“The Management Discussion and Analysis (MD&A) section, …”

Compared to the rest of the Introduction, it suggested to make this paragraph a little bit "simpler"? Readers are required to receive a lot of information in few rows.

“By employing signaling theory…”

It could be critical to assume that readers easily understand the meaning of "signaling theory" and "impression management". Some reference could support the comprehension of the sentence concept.

“It provides new perspectives …”

This paragraph could benefit from some example, so that readers can better figure out the applicative implications at this early stage of the paper.

Again, the "impression management" concept should be a bit introduced with a reference to previous studies (e.g. by Erving Goffman). Despite, it is widely diffused in the literature, a broader audience of the paper could benefit from this deepening.

The end of the Introduction paragraph could help readers to have a short and clear picture of the following chapters of the paper.

2.1. Literature review of tone manipulation

Readers could need more information about "information game theory" and they could benefit from a clearer explanation about how it has been used in this interpretation of the management disclosure strategy.

As above, the impression management process should be explained a little bit and cited to make its use clear in the research of this research.

“However, Internet platforms may also facilitate …”

The overall idea is clear, however some example could benefit readers to figure out the applicative scenarios.

“2.2. Digital Transformation and tone manipulation”

At the beginning of the paragraph, readers could benefit from a synthetic definition of the "principal-agent problem".

Also, the "signal theory" should be a bit introduced by a little definition and some reference.

It is suggested to use some practical example to clarify the effects and the biases cited (e.g. anchoring bias or "ostrich effect").

Generally, it is suggested to make sentences shorter in order to help the readers understanding of the theoretical concepts.

In particular, it is suggested to cite some reference of the Attribute substitution theory (e.g. Tversky and Kahneman when they argued the systematic errors in judgment and decision).

“5.3 Mechanism analysis”

When quoting different statements that come from previous studies and researches, it is suggested to always also cite the source.

Many "deductions" about managers' biases and overconfidence are common sense and reasonable, but in this context it is necessary to report from which studies these theories are taken or to which data of this study they are correlated.

“6. Summary and conclusion”

The conclusion paragraph is well structured. The numbered list helps a lot in reading the well-synthesized concepts. Point 6, unlike the first five points in the conclusions, however, seems more hypothetical than a real result of the research (it seems more like an interpretation of the authors, albeit sensible). The implications also appear very clear and targeted, useful for practical applications in politics and economics.

It is suggested to divide the conclusions paragraph into three areas: the first as now with the six numbered points, then a paragraph of "implications" and a final one of "limitations and future research" (the latter with a clear point for "Different types of digital transformation" and a final point for "Strategic use of digital transformation as an optimistic signal"). In this way, the paragraph could be even more schematic and therefore easier to read.

General reviewer's comments

Overall, I found the paper highly engaging and insightful in terms of understanding a specific economic and political context in China. The authors’ arguments are clearly articulated, with a focused approach towards well-defined research objectives. The initial hypotheses are also thoroughly explained, and the conclusions are effectively aligned with these hypotheses.

The introduction does a commendable job of outlining the foundational research and theories that serve as the starting point for this work. However, it would be beneficial to consistently cite references, even for widely recognized studies, to ensure that the broader audience of the journal can easily trace the sources of information.

The statistical analysis is well-structured and meticulously detailed. However, while the conclusions are particularly well-organized—summarized into clear, listed points—it is suggested to extend this structured approach to the sections on practical implications and research opportunities. This would enhance clarity and provide a more systematic overview for readers looking for takeaways and future research directions.

Additional suggestions

Some sentences, particularly those referencing various theories, are often too lengthy, which may cause readers to lose track of the argument. I recommend using shorter, grammatically simpler sentences in these cases. For instance, the use of gerunds and inserted clauses can make it difficult to follow the authors' reasoning in such a complex context. A key point for revision should focus on the linguistic form, which I suggest should be more concise, direct, and with shorter sentences, especially when explaining critical logical transitions.

Lastly, some concepts in the introductory paragraphs are slightly redundant. While this can help readers better grasp the authors' hypotheses, it may be worth considering where to reduce redundancy and where to leave it for clarity.

Reviewer #3: This paper is a valuable research topic that contributes to a deeper understanding of the intrinsic motivations and mechanisms of the impact of corporate digital transformation strategies on disclosure in the digital economy. The authors have also conducted a relatively sufficient empirical test, with a reasonable research design and rigorous analytical methods, which can effectively answer the research questions raised. The article has a novel perspective, smooth theoretical logic, and is a good innovation and supplement to the existing literature. However, some places that can be further optimized are also found, and the following modifications are summarized for the authors' reference.

1.The structure outlined in the final paragraph of the introduction does not fully align with the content of the subsequent chapters. It is recommended that the author revise this section to better reflect the actual structure of the paper.

2.In the literature review section, focusing solely on existing research from the perspective of motivation may come across as monotonous. It could be beneficial to further organize the literature content by also exploring its economic consequences. This would provide a more comprehensive understanding of the topic and potentially uncover new insights that could enhance the overall quality of the review.

3.In the hypotheses development section, the logic behind "Innovation, being the core of digital transformation, means that risk information disclosure often involves product research and development, technological innovation, and other related information" is not entirely clear. It is suggested that the author reconsider and revise this statement for improved clarity.

4.Although it is a common measure, the authors are advised to specify the source of the digital transformation measure used in the main regression and to provide a list of the keywords used in the textual analysis in the appendix.

6. PLOS authors have the option to publish the peer review history of their article (what does this mean? ). If published, this will include your full peer review and any attached files.

**Do you want your identity to be public for this peer review?** For information about this choice, including consent withdrawal, please see our Privacy Policy .

Reviewer #1: No

Reviewer #2: **Yes: ** Marco Camilli

Reviewer #3: No

---

## [Author Response · Author response to Decision Letter 1]

5 Nov 2024

We are very grateful for your valuable review comments on our study " Encourage or inhibit: A study on the impact of corporate digital transformation on management's tone manipulation of information disclosure", which has significant guidance for the revision and improvement of our study and the enhancement of academic research level!

For detailed responses to reviewer and editor comments, please refer to the uploaded attachment "Responses to Reviewers".

---

## [Decision Letter · Decision Letter 1]

28 Nov 2024

PONE-D-24-36330R1Encourage or inhibit: A Study on the Impact of Corporate Digital Transformation on Management's Tone Manipulation of Information DisclosurePLOS ONE

Dear Dr. Zhang,

Thank you for submitting your manuscript to PLOS ONE. After careful consideration, we feel that it has merit but does not fully meet PLOS ONE’s publication criteria as it currently stands. Therefore, we invite you to submit a revised version of the manuscript that addresses the points raised during the review process.

We look forward to receiving your revised manuscript.

Kind regards,

Reza Rostamzadeh

Academic Editor

PLOS ONE

Journal Requirements:

Reviewers' comments:

Reviewer's Responses to Questions

**Comments to the Author**

1. If the authors have adequately addressed your comments raised in a previous round of review and you feel that this manuscript is now acceptable for publication, you may indicate that here to bypass the “Comments to the Author” section, enter your conflict of interest statement in the “Confidential to Editor” section, and submit your "Accept" recommendation.

Reviewer #2: (No Response)

Reviewer #3: All comments have been addressed

2. Is the manuscript technically sound, and do the data support the conclusions?

Reviewer #2: Yes

Reviewer #3: Yes

3. Has the statistical analysis been performed appropriately and rigorously? 

Reviewer #2: N/A

Reviewer #3: Yes

4. Have the authors made all data underlying the findings in their manuscript fully available?

Reviewer #2: Yes

Reviewer #3: Yes

5. Is the manuscript presented in an intelligible fashion and written in standard English?

Reviewer #2: (No Response)

Reviewer #3: Yes

6. Review Comments to the Author

Reviewer #2: General Observations

1. Novelty & Relevance: The paper contributes to the literature by exploring tone manipulation in the context of digital transformation, focusing on Chinese A-share companies. This is an underexplored area and the study is relevant and timely.

2. Structure & Clarity: The structure is logical and follows standard academic conventions (Introduction, Literature Review, Methodology, Results, Discussion, Conclusion). However, some sections could benefit from better organization and brevity.

3. Example well-structured and clear writing: For all readers (highly specialized or naïve) opening pages of the paper effectively describe the context in which expectations around digital transformation influence management to overestimate its benefits, leading to highly positive information disclosures directed at shareholders.

Key Areas of Improvement

1. Words, language and style

Run-on sentences are a recurrent issue: the paper contains many overly long sentences with multiple clauses that hinder comprehension.

This is an example: "Innovation, being the core of digital transformation, means that risk information disclosure often involves product research and development, technological innovation, and other related information, although enhancing the redundancy of risk disclosure can increase firms' information transparency and reduce firms' agency costs..."

Suggestion: Split into two sentences for clarity.

Here another example: "Managers may exploit this by overstating the benefits and downplaying the challenges of digital transformation through tone manipulation, aiming to elicit market reactions that benefit their stock prices."

Suggestion: Break into two shorter, precise sentences.

This typo of phrases: "Digital transformation positively affects various aspects of enterprises, including production costs, business performance, competitive advantage, and innovation" are repeated in slightly different forms.

Suggestion: Eliminate redundancy and simplify phrasing.

Inconsistent Vocabulary: Terms like "media optimism" and "media slant" are used interchangeably, potentially confusing readers.

Suggestion: Standardize terminology throughout the text.

2. One missing references

The key theory like impression management (line 102) lacks foundational references.

Recommendation: Add citations to foundational works (e.g., Goffman).

3. Empirical Analysis

Clarity:

While the methodology is comprehensive, the presentation is overly technical in some sections. For instance, the explanation of "Abtone" (line 474) could be simplified for clarity.

Suggestion: Use clear transitions between statistical details and their implications to improve readability.

4. Practical Implications

The paper provides meaningful insights for investors and regulators, but these are scattered.

A solution could be to consolidate these into a dedicated "Practical Implications" section.

5. Redundancy

Several points are repeated across sections (e.g., the benefits and risks of digital transformation). This could make the paper appear unnecessarily lengthy.

Please, take into consideration to consolidate repeated points, particularly in the introduction and literature review.

Final considerations

This revised version of the paper addresses several critical issues of the first contribution. It sounds much more readable for a wider audience. It is still suggested to concise several long sentences, to reference all theories and concepts (to support readers in literature review), and also to reduce to the minimum redundant thoughts. Using direct, clear, and concise sentences in a dry, brief communication style can help readers become more interested and engaged in the content.

Reviewer #3: I have carefully reviewed your revised manuscript, and I am very delighted to see that you have made careful revisions and improvements to my previous comments and suggestions. I believe that your article is informative, logical and clear, and has reached the publication standard of the journal. So I recommend for publishing the article!

7. PLOS authors have the option to publish the peer review history of their article (what does this mean? ). If published, this will include your full peer review and any attached files.

**Do you want your identity to be public for this peer review?** For information about this choice, including consent withdrawal, please see our Privacy Policy .

Reviewer #2: **Yes: ** Marco Camilli

Reviewer #3: No

---

## [Author Response · Author response to Decision Letter 2]

19 Dec 2024

On behalf of my co-authors, we thank you very much for giving us an opportunity to revise our manuscript, and we also appreciate reviewers very much for their positive and constructive comments and suggestions on our manuscript.

Following the comments and suggestions received, we have made modifications. which are summarized below:

1. Redundancy. We had some unnecessary repetitive descriptions in various sections. We thought more detail would be better, but it backfired, making the article too verbose. We modified unnecessary repetitions in the text. This includes content about the benefits of digital transformation, the purpose of tone manipulation and the current state of digital transformation risk and so on.

2. References. We added references to related literature based on the existing studies. These references can help readers better understand the theories mentioned in this paper

3. Variable definition. We revised the variable definition section. This makes the content more logical and easier for readers to understand.

4. Chapter structure. The structure of Chapter Six is indeed unclear. We added three subheadings in Chapter Six: 6.1 Conclusions; 6.2 Practical implications; 6.3 Limitations and prospects. These additions are based on the research content. They make it easier for readers to understand the main points of this section.

The specific can be seen in the attached file "Response to Reviewers".

Once again, thank you very much for your comments and suggestions. And we hope that the revised manuscript can be accepted by PLOS ONE. If further revision is necessary, please contact me.

Thank you and best regards.

---

## [Editor Report · Decision Letter 2]

30 Dec 2024

Encourage or inhibit: A Study on the Impact of Corporate Digital Transformation on Management's Tone Manipulation of Information Disclosure

PONE-D-24-36330R2

Dear Dr. Zhang,

We’re pleased to inform you that your manuscript has been judged scientifically suitable for publication and will be formally accepted for publication once it meets all outstanding technical requirements.

Kind regards,

Reza Rostamzadeh

Academic Editor

PLOS ONE
---

## [Editor Report · Acceptance letter]

PONE-D-24-36330R2

PLOS ONE

Dear Dr. Zhang,

I'm pleased to inform you that your manuscript has been deemed suitable for publication in PLOS ONE. Congratulations! Your manuscript is now being handed over to our production team.

Kind regards,

on behalf of

Dr. Reza Rostamzadeh

Academic Editor

PLOS ONE